

# Studying suicide using proxy-based data: reliability and validity of a short version scale for measuring quality of life in rural China

Huiming He[1,2], Qiqing Mo[1], Xinyu Bai[1], Xinguang Chen[3], Cunxian Jia[4], Liang Zhou[5] and Zhenyu Ma[1]

[1] School of Public Health, Guangxi Medical University, Nanning, Guangxi, China
[2] Institute of Parasitic Disease Control and Prevention, Guangxi Zhuang Autonomous Region Center for Disease Control and Prevention, Nanning, Guangxi, China
[3] University of Florida, Florida, United States of America
[4] School of Public Health, Shandong University, Jinan, Shandong, China
[5] The Affiliated Brain Hospital of Guangzhou Medical University, Guangzhou, Guangdong, China

## ABSTRACT

**Background:** To evaluate the reliability and validity of the short version six-item Quality of Life Scale (QOLS-6) and the consistency of subject-proxy data in a case-control psychological autopsy study on elderly suicide in rural China.
**Methods:** A two-stage stratified cluster sampling method was used to select research sites. We used self-administered questionnaires to collect proxy-based information from informants and subject-based information from living comparisons.
**Results:** A total of 242 pairs of suicide cases and living comparisons were selected in our research. Subject-proxy consistency for QOLS-6 was good (Intraclass correlation coefficient (ICC) was 0.688) in living controls. Good internal consistency of QOLS-6 was validated by Cronbach's α being greater than 0.6 among suicide cases and living comparisons. The mean scores of quality of life were lower among suicide cases than living controls. Quality of life was negatively correlated with depression, loneliness, hopelessness, impulsiveness and stressful life events, while it was positively correlated with activities of daily living and family function.
**Conclusions:** QOLS-6 has good reliability and validity, which can be used for assessing quality of life among Chinese rural older adults. It is shorter and easier than any other scale for measuring quality of life and can be used as a screening tool in future studies.

# INTRODUCTION

The World Health Organization defines quality of life as an individual's perception of their position in life in the context of their culture and value systems. It is affected in a complex way by the person's physic, psychology, level of independence, social relationships, and environment (*WHOQOL-Group, 1993*).

Corresponding authors
Liang Zhou,
Liangzhou_csu@vip.163.com
Zhenyu Ma,
ma_zhenyu@gxmu.edu.cn

Quality of life includes a person's physical health, psychological state, and social relationships. According to the definition of healthy, quality of life can be considered a reflection of the health status. Some researchers found that quality of life is one of the critical factors leading to suicidal behavior (*Balazs et al., 2018*).

As the population aging process accelerates, we should pay more attention to quality of life among the elderly to ensure that their life is worth living. The elderly's income may be reduced due to loss of work, and the living environment also undergoes certain changes. With advancing age, the elderly begin to develop chronic diseases, face reduced income, and lose some connection with their networks. These will inevitably cause declines in their quality of life.

The population aged 60 or older reached 900 million worldwide (*WHO, 2016*). In China, the population aged 60 or older accounted for 17.9% of the general population and those aged 65 or older accounted for 11.9% in 2018 (*National Bureau of Statistics of China, 2019*). Moreover, suicide rates are highest in both old men and women in almost all regions of the world (*WHO, 2018*). It was estimated that the average suicide rate among the elderly was 6.2-fold higher than that among the population under 65 (*Zhong, Chiu & Conwell, 2016*). Suicide rates among Chinese rural older adults are the highest compared to all age groups; however, little is known about quality of life in this rapidly growing vulnerable population.

There are many well-developed scales used to measure quality of life, such as WHOQOL-100, WHOQOL-BREF, EQ-5D, SF-36, and so on (*Makovski et al., 2019*). However, no studies on the reliability and validity of these scales were conducted in elderly suicide cases. *Phillips et al. (2002)* developed the Quality of Life scale (QOLS-6) based on their psychological autopsy studies among the general population in 2002. As a short version measurement, this scale is easier and more convenient to carry out in suicide research especially among the elderly. Thus, the purpose of this study is to evaluate the reliability and validity of this scale. As far as we know, this study is the first to evaluate QOLS-6 in several sequential periods in a large sample of rural older adults. In short, as the population aging rate and the number of elderly people increase rapidly, the rural elderly have a higher suicide rate, and the quality of life of the elderly is receiving increasing attention. Therefore, this is a very useful study on the quality of life for suicidal elderly. This scale has practical significance for clinical work. We can assess the person's quality of life quickly through it and quality of life can be seen as a reflection of personal health. So it means we can promote health and suicide prevention through this scale.

## MATERIALS & METHODS

### Participants and procedures

A two-stage stratified cluster sampling was used to select research sites. Firstly, according to the GDP *per capita* of 31 provinces in mainland China (*National Bureau of Statistics of China, 2015*), 3 provinces—Shandong, Hunan, and Guangxi, were randomly selected from the top 10, 11–20 and 21–31 provinces in terms of GDP. Secondly, counties of each province were divided into three levels according to economic status and 12 counties were randomly selected. Suicide cases aged 60 or older were collected continuously based on the

death certificate system in each county. According to age (±3 years old), sex, and living location of suicide cases, living comparisons were 1:1 matched. When a suicide case was selected, trained investigators would list and numbered suitable living older adults, and then a living comparison was selected randomly from the list by a computer program. We selected two informants for each suicide case and living comparison, respectively. The first informant (*e.g.*, a next-to-kin) was always the person who lived with a suicide case or living comparison, whereas the second informant (*e.g.*, a friend, a neighbor or a relative) usually didn't lived with them. Interviews with informants of living comparisons were conducted as soon as comparisons and their informants were identified, while interviews with informants of suicide cases were conducted in a schedule of 2–6 months after death. Each informant was interviewed separately by one trained interviewer. The average interview time was 90 min. The content of the interview with the informants was to reflect the situation of the target person (suicide case or living control) rather than the informants. The information of a suicide case was obtained from the description of two corresponding informants. Besides, the information of a living control was obtained by himself as well as the corresponding informants. Thus, the information obtained from the living control themselves was also called the gold standard. Questionnaires are almost the same between informants and living controls themselves, the questionnaire of informants is interviewed for information of the target person (a suicide case or a living comparison) but not informants. A total of 242 suicides and 242 living controls were investigated in this study. Informed consent was obtained from participants in this study by signing the informed consent form prior to the interview. The participants of suicide case group were the informants of suicide cases, but those of living control group were living controls and their informants. This study was approved by the Human Research Ethics Committees of Shandong University (Ethical Application Ref: 20150306-1), Central South University (Ethical Application Ref: CTXY-130041-1), and Guangxi Medical University (Ethical Application Ref: 20150146).

There were some discrepancies and differences from two informants for a target person, thus the information needed to be integrated. For the demographic characteristics, the information provided by the first informant was more reliable than those from second informant. If the target person has behaviors that increase the risk of suicide, this answer of informant should be adopted. Because when one of the two informants observed targeting behavior, it may exist.

## Measurements

### Demographic characteristics

Demographic data such as sex, age, education level, family income, marital status, employment status, and living arrangements were collected. Education level included below primary school, primary school and above primary school. Based on 33 and 66 percentiles, family annual income in Chinese yuan (CNY) was divided into three groups: <3,600 yuan, 3,600–10,000 yuan and >10,000 yuan. For marital status, people who were married and living with the spouse or cohabiting were classified as "stable marital status", while people with another marital status (including single, divorced, widowed,

married but living apart) were classified as "unstable marital status". Being left-behind is defined as all adult family members who have been away from home for at least 10 months and had visited the elderly no more than twice in the nearly 12 months prior to death/investigation.

### Quality of life

Quality of Life scale (QOLS-6) (*Phillips et al., 2002*) developed by Phillips was adopted, in which six items in question form were used to collect the quality of life of the target person in the last month before the suicide or investigation. It includes physical health, psychological health, economic status, work, family relationships and relationships with others. On each item, participants were asked to indicate the status using a five-point scale: one = very poor, two = poor, three = fair, four = good and five = excellent. The total score ranges from 6 to 30, higher scores indicate higher living quality.

### Other related factors

Geriatric Depression Scale (GDS) (*Chan, 1996*; *Yesavage et al., 1982*), Chinese Revision of Short-form of the UCLA Loneliness Scale (ULS-6) (*Hays & DiMatteo, 1987*; *Zhou et al., 2012*), short Beck Hopeless Scale (BHS-4) (*Beck & Weissman, 1974*; *Liu et al., 2011*), Barratt Impulsiveness Scale (BIS-11) (*Li et al., 2011*; *Patton, Stanford & Barratt, 1995*), Life Events Scale for the Elderly (LESE) (*Xiao & Xu, 2007*), and Duke Social Support Index (DSSI) (*Koenig et al., 1993*; *Mao et al., 2015*), Activity of Daily Living Scale (ADL) (*Lawton & Brody, 1969*) and the family Adaptive, Partnership, Growth, Affection and Resolve Scale (family APGAR) (*Smilkstein, 1978*) were also used in this study to measure related suicide risk factors. All of these measurement tools had been widely applied in suicide research to evaluate people's physical, psychological, and social well-being and showed good reliability and validity among Chinese rural suicide elderly (*Mo et al., 2019*; *Niu et al., 2018a*; *Niu et al., 2018b*).

## Statistical analysis

Descriptive analysis, McNemar's test, paired t-test, Wilcoxon signed-rank test were used to compare demographic characteristics. Intraclass correlation coefficient (ICC) was used to evaluate the agreement between proxy-based data from informants and subject-based data from living controls (gold standard). Exploratory factor analysis was used to analyze structural validity of QOLS-6. Concurrent validity of QOLS-6 were examined by using Spearman's correlation analysis between QOLS-6 and other related scales.

Statistical analysis was performed by using SPSS 23.0 for Windows (SPSS Inc, Chicago, IL, USA). All reported $P$ values were two-sided, and $P$ values $< 0.05$ were considered statistically significant.

## RESULTS

### Demographic characteristics and other scales

A total of 242 elderly suicide cases and 242 living comparisons were investigated. As shown in Table 1, no significant differences were found in education, family annual income. Suicide cases were significantly more likely to have unstable marital status,

**Table 1 Demographic characteristics and comparisons of suicide cases and living controls.**

| Demographic characteristics | Suicide cases (n = 242) | Living controls (n = 242) | $\chi^2/t/(Z/W_+)$ | P |
|---|---|---|---|---|
| **Sex** | | | — | — |
| Male | 135 (55.79) | 135 (55.79) | | |
| Female | 107 (44.21) | 107 (44.21) | | |
| **Age groups** | | | 0.913 | 0.633 |
| 60–69 | 73 (30.17) | 70 (28.93) | | |
| 70–79 | 100 (41.32) | 110 (45.45) | | |
| ≥80 | 69 (28.51) | 62 (25.62) | | |
| **Education** | | | 3.199 | 0.362 |
| Below primary school | 111 (45.87) | 96 (39.67) | | |
| Primary school | 105 (43.39) | 116 (47.93) | | |
| Above primary school | 26 (10.74) | 30 (12.40) | | |
| **Employment** | | | 9.149 | 0.027 |
| Employed | 41 (16.94) | 61 (25.21) | | |
| Unemployed | 194 (80.17) | 167 (69.01) | | |
| Retired | 7 (2.89) | 14 (5.78) | | |
| **Family annual income (CNY)** | | | 2.122 | 0.547 |
| <3,600 | 88 (36.36) | 74 (30.58) | | |
| 3,600–10,000 | 88 (36.36) | 98 (40.50) | | |
| >10,000 | 66 (27.28) | 70 (28.92) | | |
| **Marital status** | | | 21.240 | <0.001 |
| Stable marital status | 122 (50.41) | 170 (70.25) | | |
| Unstable marital status | 120 (49.59) | 72 (29.75) | | |
| **Suffering from serious or chronic diseases** | | | 17.582 | <0.001 |
| Yes | 202 (83.47) | 161 (66.53) | | |
| No | 40 (16.53) | 81 (33.47) | | |
| **Living alone** | | | 9.924 | 0.001 |
| Yes | 64 (26.45) | 35 (14.46) | | |
| No | 178 (73.55) | 207 (85.54) | | |
| **Being left behind** | | | 4.500 | 0.033 |
| Yes | 41 (16.94) | 25 (10.33) | | |
| No | 201 (83.06) | 217 (89.67) | | |
| Age, mean (SD) | 74.43 (8.22) | 74.05 (8.16) | 2.080 | 0.039 |
| GDS, mean (SD) | 21.41 (5.95) | 9.22 (6.42) | 21.604 | <0.001 |
| ULS-6, mean (SD) | 15.55 (5.00) | 10.99 (4.01) | 11.855 | <0.001 |
| BHS, mean (SD) | 14.55 (2.54) | 9.66 (2.72) | 20.049 | <0.001 |
| BIS-11, mean (SD) | 98.79 (16.63) | 86.91 (15.18) | 8.888 | <0.001 |
| DSSI, mean (SD) | 22.88 (5.98) | 27.47 (6.82) | −8.843 | <0.001 |
| ADL, mean (SD) | 46.26 (12.49) | 52.47 (6.03) | −7.148 | <0.001 |
| APGAR, mean (SD) | 4.50 (3.39) | 7.02 (2.79) | −9.487 | <0.001 |
| No. life events, median (QR) | 5.00 (4.00) | 3.00 (4.25) | -5.666/6456.50 | <0.001 |

Note:
GDS, Geriatric Depression Scale; ULS-6, University of California Los Angeles Loneliness Scale-6; BHS, Beck's Hopelessness Scale; BIS-11, Barratt Impulsiveness Scale; DSSI, Duke Social Support Index; ADL, Activity of Daily Living Scale; APGAR, family Adaptive, Partnership, Growth, Affection and Resolve Scale.

**Table 2 Consistency test between living comparisons and proxy information.**

| Items | Comparisons and first informants | | Comparisons and second informants | | Comparisons and integrated information | |
|---|---|---|---|---|---|---|
| | ICC | *P* | ICC | *P* | ICC | *P* |
| 1. How was your physical health in the last month? | 0.571 | <0.001 | 0.505 | <0.001 | 0.621 | <0.001 |
| 2. How was your psychological health in the last month? | 0.526 | <0.001 | 0.362 | <0.001 | 0.492 | <0.001 |
| 3. How was your economic status in the last month? | 0.546 | <0.001 | 0.491 | <0.001 | 0.571 | <0.001 |
| 4. How was your work (study or farm work) in the last month? | 0.541 | <0.001 | 0.505 | <0.001 | 0.591 | <0.001 |
| 5. How were your relationships with your family in the last month? | 0.418 | <0.001 | 0.297 | 0.003 | 0.318 | 0.002 |
| 6. How were your relationships with others in the last month? | 0.390 | <0.001 | 0.462 | <0.001 | 0.468 | <0.001 |
| Total scores | 0.653 | <0.001 | 0.621 | <0.001 | 0.688 | <0.001 |

unemployed, living alone, being left-behind and suffering from serious or chronic diseases. Also, suicide cases show higher scores in depression, loneliness, hopelessness, impulsiveness and life events, lower scores in social support, activity of daily living and family function than living controls ($P < 0.001$).

## Reliability

Cronbach's α of suicide cases and living controls was 0.641 and 0.782, respectively, which indicated that the internal consistency of QOLS-6 was good.

The consistency between gold standards and proxy information among living controls is shown in Table 2. ICC values of living comparisons between the first informant, the second informant, and integrated information are 0.653, 0.621, and 0.688, respectively.

## Construct validity

Data on suicide cases and living controls of QOLS-6 were performed using the Kaiser–Meyer–Olkin (KMO) measure of sampling adequacy and Bartlett's spherical test. The results of this study were KMO = 0.659, $\chi^2 = 365.350$, $P < 0.001$ for suicide victims and KMO = 0.737, $\chi^2 = 449.878$, $P < 0.001$ for living comparisons. This study was appropriate for exploratory factor analysis to discuss the factors structure, because both groups have KMO > 0.600 and $P < 0.05$.

Two common factors were extracted using exploratory factor analysis. The cumulative variance percentage of these two factors was 65.847% for suicide victims and 69.032% for living comparisons. Scree plots of exploratory factor analysis of suicide cases and living controls are shown in Figs. 1 and 2, respectively. The rotated component matrix is shown in Table 3. Factor 1 mainly explains Items 1 to 4, which is an internal self-related component. Factor 2 mainly explains Items 5 and 6, which is an external relationship-related component.

## Concurrent validity

The concurrent validity of QOLS-6 were examined using Spearman's correlation coefficients between scores of QOLS-6 and other related scales. As shown in Table 4,

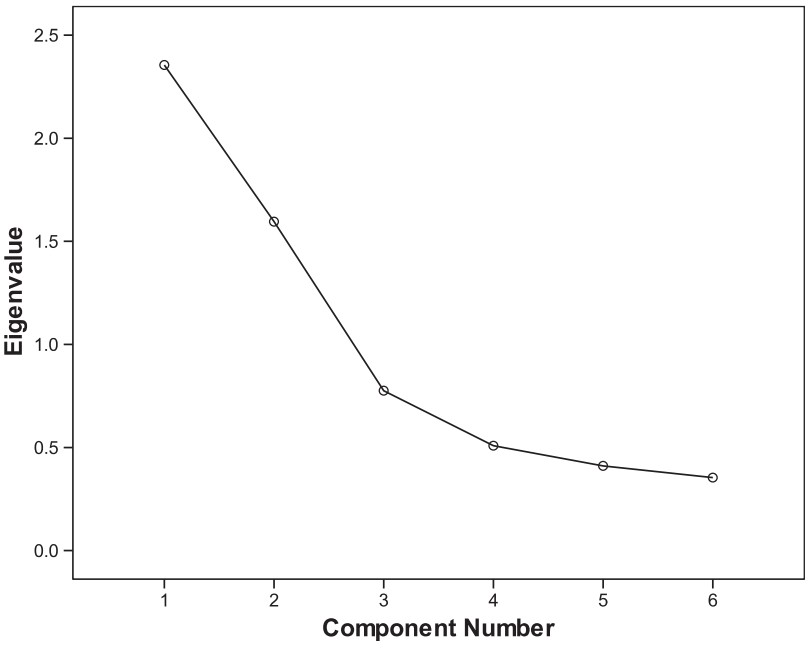

**Figure 1 Scree plot of exploratory factor analysis of suicide cases.**

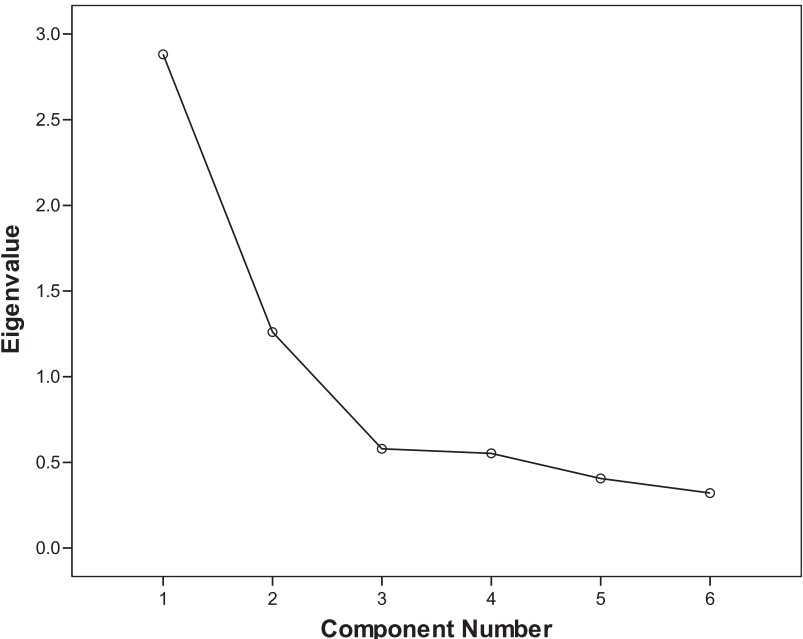

**Figure 2 Scree plot of exploratory factor analysis of living controls.**

quality of life is positively correlated with activities of daily living and family function and is negatively correlated with depression, loneliness, hopelessness, impulsiveness, and life events both among suicide cases and living controls.

**Table 3 Rotated component matrix for each item in the QOLS-6.**

| Items | Components of suicide cases | | Components of living controls | |
|---|---|---|---|---|
| | Factor 1 | Factor 2 | Factor 1 | Factor 2 |
| Item 1 | 0.833 | −0.157 | 0.846 | 0.001 |
| Item 2 | 0.816 | 0.047 | 0.796 | 0.172 |
| Item 3 | 0.591 | 0.243 | 0.696 | 0.274 |
| Item 4 | 0.798 | −0.014 | 0.749 | 0.157 |
| Item 5 | −0.006 | 0.892 | 0.151 | 0.883 |
| Item 6 | 0.048 | 0.849 | 0.158 | 0.888 |

**Table 4 Correlation coefficient between the QOLS-6 and other relevant scales.**

| Scales | QOLS-6 score | | | |
|---|---|---|---|---|
| | Suicide cases ($n = 242$) | | Living controls ($n = 242$) | |
| | r | P | r | P |
| GDS | −0.568 | <0.001 | −0.574 | <0.001 |
| ULS-6 | −0.344 | <0.001 | −0.382 | <0.001 |
| BHS-4 | −0.499 | <0.001 | −0.580 | <0.001 |
| BIS-11 | −0.287 | <0.001 | −0.482 | <0.001 |
| LESE | −0.444 | <0.001 | −0.572 | <0.001 |
| DSSI | 0.072 | 0.263 | 0.426 | <0.001 |
| ADL | 0.329 | <0.001 | 0.303 | <0.001 |
| APGAR | 0.188 | 0.003 | 0.359 | <0.001 |

Note:
QOLS-6, Quality of Life Scale; GDS, Geriatric Depression Scale; ULS-6, University of California Los Angeles Loneliness Scale-6; BHS, Beck's Hopelessness Scale; BIS-11, Barratt Impulsiveness Scale; LESE, Life Event Scale for the Elderly; DSSI, Duke Social Support Index; ADL, Activity of Daily Living Scale; APGAR, family Adaptive, Partnership, Growth, Affection and Resolve Scale.

## Quality of life in suicide cases and living controls

As shown in Table 5, compared with living controls, suicide cases have a lower score, which indicates a poorer quality of life among suicide cases than living controls ($P < 0.001$).

## DISCUSSION

This study was designed to assess the reliability and validity of QOLS-6 through psychological autopsy among elderly suicide cases and controls in rural China. The scale reported in this study provides a useful and practical tool much needed for research on older adults. It evaluates quality of life in three dimensions: physical, mental, and social well-being. The result showed that QOLS-6 had good reliability and validity among Chinese rural older adults. This instrument is simple to use and well accepted by rural residents or others with a minimum of primary school education.

It is important to use a reliable scale to measure quality of life for suicide prevention. Poor quality of life is associated with increased risk of suicide (*He et al., 2017*). By contrasting

**Table 5 Comparison of quality of life between suicide cases and living controls.**

| Items | Suicide cases ($n = 242$) Mean ± SD | Living controls ($n = 242$) Mean ± SD | t | P |
|---|---|---|---|---|
| Item 1 | 1.98 ± 0.946 | 2.99 ± 0.876 | −11.973 | <0.001 |
| Item 2 | 2.09 ± 0.881 | 3.36 ± 0.751 | −16.963 | <0.001 |
| Item 3 | 2.62 ± 0.754 | 2.90 ± 0.772 | −4.020 | <0.001 |
| Item 4 | 2.14 ± 0.836 | 2.83 ± 0.815 | −9.441 | <0.001 |
| Item 5 | 3.19 ± 0.946 | 3.74 ± 0.611 | −8.020 | <0.001 |
| Item 6 | 3.39 ± 0.727 | 3.68 ± 0.634 | −5.299 | <0.001 |
| Total score | 15.40 ± 3.062 | 19.50 ± 3.108 | −14.613 | <0.001 |

the proxy data of the suicide group and the control group, QOLS-6 can play a better role in suicide prevention. In Liu Yao's research (*Liu et al., 2014*), the score of quality of life among college students who attempted suicide was 58.41 (the total score was 100), and the average score of those without suicide attempts was 66.62. Both scores are slightly higher than those in our study. There are two inferences about this: different age groups have different quality of life, and the person who committed suicide and the one who attempted suicide may have different quality of life. Quality of life was lower among suicide cases than living controls (*Li et al., 2005*), indicating that poor quality of life should be considered a risk factor for elderly suicide cases. Quality of life showed the potential to identify the elderly who may be at risk of suicide, but the mechanism of effect on suicide also needs to be carried out in follow-up research. Therefore, this scale has practical significance for clinical work.

The short version of the QOL scale has good reliability, and proxy data are reliable for evaluating the quality of life of the target population. The ICC between gold standard and informants is greater than 0.6, which indicates that the quality of life obtained through proxy data is reliable. This result is consistent with other studies (*Wang et al., 2007*). ICC values range from 0 to 1, with 0 being untrusted and one fully credible, and are evaluated based on the following benchmarks: poor (<0.40), fair (0.40–0.59), good (0.60–0.74) and excellent (>0.75) (*Cicchetti, 1994*). Good internal consistency of QOLS-6 was validated by that two Cronbach's alpha were both greater than 0.6 among suicide cases and living comparisons, which was also consistent with previous studies (*Tian et al., 2013*).

Moreover, the scale has good construct validity. Two factors were found for the suicide and control groups, which could be defined as the internal and external components, respectively. The internal components mainly reflect the physical, psychological, economic, and working conditions, while the external components are mainly related to relationships with family or others. The items which the factor loading concentrated on were consistent both in suicide and comparison cases, which indicated that the scale's structural validity was acceptable.

The scale has good concurrent validity. Older adults with depression, loneliness, hopelessness, impulsiveness, or low family function have a poor quality of life. Previous

studies indicate that depression is associated with quality of life, and people with good quality of life suffer less from depressive symptoms (*Zhang et al., 2019*). Similarly, in this study, quality of life is negatively correlated with depression in older adults. The number of life events is negatively correlated with quality of life (*Tian et al., 2013*). Some studies indicate that increased loneliness is associated with reduced quality of life (*Arslantaş et al., 2015*), which is consistent with the results of our study. Meanwhile, previous studies (*İzci et al., 2018*) showed that both hopelessness and impulsiveness reduce quality of life. Similarly, our study shows that quality of life is negatively correlated with hopelessness and impulsiveness. Moreover, our study shows that people who are able to do living activities have good quality of life, which is consistent with previous reports (*Yang et al., 2017*). The APGAR is positively correlated with the QOLS-6 (*Lu et al., 2017*). Good family function could improve the elderly's quality of life. Therefore, the quality of life scale also shows good concurrent validity according to the correlation between the above mentioned factors.

This scale has practical significance for clinical work. Through above analysis, we can know that quality of life is related to suicide and various other mental health problems (Such as depression, etc.). Therefore, we can use it to predict and prevent the risk of suicide and other mental health problems.

There are some limitations to this study. There is information bias in both the suicide cases and living controls, while the case-control design can minimize the impact of information bias. In this study, we used living comparisons as the gold standard to assess the validity of their informants' reports. Informants in suicide group may be in grief and reported information may be affected. This study adjusted and matched the age, sex, and living location to identify potential risk factors, which meant that we could not consider the potential interaction between these factors and other potential risk factors. Men and women are very different in their communications and interpersonal interactions. Thus, we may miss a key potential element when not comparing results for men and women. In addition, with the rural elderly as the study objects in this study, further research should be carried out in other groups, and more other risk factors should be considered.

## CONCLUSIONS

In conclusion, QOLS-6 scale has good reliability and validity for assessing quality of life among older adults in China. It is shorter and easier than any other scale for measuring quality of life and could be used as a screening tool in future studies. With changes in the medical models, future research should pay more attention to mental and social factors. In addition to the components of physical and psychological health, the scale also explains quality of life from a social perspective, which is more conducive to comprehensive assessment of quality of life. As our study focuses on Chinese rural older adults, we recommend that future studies should be conducted on different age groups in various populations to further evaluate the validity and reliability of QOLS-6. Quality of life showed the potential to identify the elderly who may be at risk of suicide, but the mechanism of effect on suicide may need to be carried out in follow-up research.

### Funding

This study was conducted in Hunan, Shandong, and Guangxi provinces in China, and was supported by the American Foundation of Suicide Prevention (Grant No. SRG-0-169-12), the Science and Technology Plan Project of Guangdong Province (No. 2019B030316001), and the Natural Science Foundation of Guangxi Zhuang Autonomous Region (Grant No. 2014GXNSFBA118163). The funders had no role in study design, data collection and analysis, decision to publish, or preparation of the manuscript.

### Grant Disclosures

The following grant information was disclosed by the authors:
American Foundation of Suicide Prevention: SRG-0-169-12.
Science and Technology Plan Project of Guangdong Province: 2019B030316001.
Natural Science Foundation of Guangxi Zhuang Autonomous Region: 2014GXNSFBA118163.

### Competing Interests

The authors declare that they have no competing interests.

### Author Contributions

- Huiming He analyzed the data, prepared figures and/or tables, authored or reviewed drafts of the paper, and approved the final draft.
- Qiqing Mo analyzed the data, authored or reviewed drafts of the paper, and approved the final draft.
- Xinyu Bai analyzed the data, authored or reviewed drafts of the paper, and approved the final draft.
- Xinguang Chen analyzed the data, authored or reviewed drafts of the paper, and approved the final draft.
- Cunxian Jia performed the experiments, authored or reviewed drafts of the paper, and approved the final draft.
- Liang Zhou conceived and designed the experiments, authored or reviewed drafts of the paper, and approved the final draft.
- Zhenyu Ma performed the experiments, authored or reviewed drafts of the paper, and approved the final draft.

### Human Ethics

The following information was supplied relating to ethical approvals (*i.e.*, approving body and any reference numbers):

This study was approved by the Human Research Ethics Committees of Shandong University, Central South University, and Guangxi Medical University. Written informed consent has been obtained from participants in the living comparison group, and all informants of both suicide cases and living comparisons.

## Data Availability

The raw data is available in the Supplementary File.

## Supplemental Information

Supplemental information for this article can be found online at http://dx.doi.org/10.7717/peerj.12396#supplemental-information.

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
