# Peer review of "Studying suicide using proxy-based data: reliability and validity of a short version scale for measuring quality of life in rural China"

_PeerJ, doi:10.7717/peerj.12396_

## Round 0.1 · original submission · Major Revisions

This is an interesting study but the current version needs substantial revisions. Please consider these comments and revise it accordingly.

Reviewer 1 ·

Basic reporting

This is a very well-written, succinct paper regarding the use of the QOLS-6 involving 242 elderly suicides and matched controls in rural China, with informants for both and direct interviews for the controls. The study has been well-designed and supports between group comparisons--suicide-informants v. control-informants, and controls v. control-informants. Its results provide confidence in the use of this measure!

There is nothing specific in the current manuscript that I would critique, in terms of methods, data acquisition, analysis, or interpretation. However, the authors miss a key potential element when not comparing results for men and women, to assure the same high level of confidence that is evident with the overall results. While one might expect findings to be similar, we know that men and women are very different in their communications and interpersonal interactions, and it would appear that the sample size would support exploratory sub-analyses.

The use of English in this manuscript is quite good. I do see one error on line 155...presumably, "wildly" is meant to be "widely."

Experimental design

As above

Validity of the findings

As above

Additional comments

As above

·

Basic reporting

1. The title of this article is ambiguous. Since the authors have addressed that “the purpose of this study is to evaluate the reliability and validity of this scale” in lines 90-91, “studying suicide using proxy-based data” in the title seems unnecessary.
2. The rationale of conducting this study is not well justified. What benefit or practical significance (such as suicide prevention) could this scale bring to the clinical work or real-world situation? Please illustrate in the Introduction part.
3. In lines 83-84, the sentence “however, little is known about suicide risks in this rapidly growing vulnerable population” seems irrelevant because this study didn’t investigate the risk factors of suicide.
4. In line 79, there is format error in the reference “(China, 2019)”.
5. In lines 99-100, a citation is needed for the sentence “according to the GDP per capita of 31 provinces in mainland China”.
6. In lines 113-114, does the “average interview time of 90 minutes” refer to a requirement to investigators/interviewers or a calculated average of actual interview time?
7. In lines 115-117, it would be better understandable if the authors specify which parts of the questionnaires were obtained from the informants and which parts of the questionnaires were obtained from the living controls themselves. Were the sources of demographic information and psychological assessments different?
8. In line 118, the sentence “the living control was also called the gold standard” was vague. Does the gold standard refer to the information obtained from the living controls themselves or the information obtained from the informants of living controls? Please specify.
9. In line 132, the international abbreviation for Chinese yuan is CNY, not RMB.
10. In lines 148-157, the authors should provide all the references of original English versions and validated Chinese versions of all the scales used in this study.
11. It would be better if the authors could provide the scree plots of exploratory factor analysis.
12. The full English name of acronyms should be spelled out when the acronym appeared the first time no matter in Abstract or main text part.
13. In lines 222- 224, how was the sentence “both scores are slightly higher than those in our study” concluded since the total score in Liu’s study is 100 while the total score in this study is 30?
14. It would be better and more readable if the authors could reorganize Table 1 following the attached format. (I have uploaded an attached PDF containing the suggested format of Table 1)

Experimental design

1. The target population in this study is inconsistent. For example, in lines 55-56, the sentence “QOLS-6 … can be used for assessing quality of life among Chinese rural older adults”; in line 86-87, the sentence “however, no studies on the reliability and validity of these scales were conducted in elderly suicide cases”. Is the target population in this study older adults or elderly suicide cases?
Similar situation happened in lines 213-217, the sentence “The scale … provides a useful and practical tool much needed for research on older adults” and the sentence “QOLS-6 has good reliability and validity among Chinese rural elderly suicide cases using proxy-based data”.
Please make the target population and principle study purpose clear throughout the study.
2. In lines 119-120, please specify the participants from which the informed consent was obtained. The “participants” might differ in suicide case group and living control group.
3. In lines 163-164, the sentence “ICC is also one of the indicators for measuring and evaluating inter-observer reliability and test-retest reliability” seems unnecessary, because inter-observer reliability and test-retest reliability were not conducted in this study. Inter-observer reliability or inter-rater reliability refers to the degree of agreement among two independent interviewers/raters/trained investigators.
4. In lines 200-205, the terminologies of “convergent validity” and “discriminant validity” were not appropriate here. Convergent validity can be established if two similar constructs correspond with one another, while discriminant validity applies to two dissimilar constructs that are easily differentiated. Since QOLS-6 and all the other scales used in Table 4 assess different aspects of the target individuals, the authors could consider using the concept of “concurrent validity”, although concurrent validity and convergent validity are both evaluated by correlation coefficient.
5. In line 231, the terminologies of “external and internal reliability” is not in common use in psychometric studies. The reliability of a scale usually includes internal consistency reliability, test-retest reliability, inter-rater reliability, and parallel-form reliability. Please reorganize the methods and results of reliability accordingly.

Validity of the findings

1. In line 160, McNemar’s test rather than Chi-square test, and paired t-test rather than independent sample t-test should be applied because this is a paired sample.
2. In line 160, the authors declared that Wilcoxon signed-rank test was applied, but the statistics of Wilcoxon signed-rank test are W+ (the sum of positive ranks) and W- (the sum of negative ranks), which were not reported in Table 1.
3. In Discussion section, the authors should introduce the practical significance of this scale to clinical work, and the methodological limitations of this study.

---

## Round 0.2 · accepted · Accept

I am pleased to accept this revised manuscript.

·

Basic reporting

The authors have made adequate revisions and reasonable responses according to the previous comments. I have no further comments.

Experimental design

The authors have made adequate revisions and reasonable responses according to the previous comments. I have no further comments.

Validity of the findings

The authors have made adequate revisions and reasonable responses according to the previous comments. I have no further comments.